# Characterizing Gene-Level Adaptations in the Gut Microbiome During Viral Infections: The Role of a Fucoidan-Rich Extract

**DOI:** 10.3390/genes16070740

**Published:** 2025-06-26

**Authors:** Gissel García, Josanne Soto, Carmen Valenzuela, Raul De Jesús Cano

**Affiliations:** 1Pathology Department, Hermanos Ameijeiras Clinical and Surgical Hospital, La Habana 10400, Cuba; gisselgarcia2805@gmail.com; 2Clinical Laboratory Department, Clinical Hospital “Hermanos Ameijeiras”, La Habana 10400, Cuba; josanne.soto@infomed.sld.cue-mail.com; 3Institute of Cybernetic, Mathematics and Physics Havana University, La Habana 10400, Cuba; carmenvalenzuelasilva@gmail.com; 4Biological Sciences Department, California Polytechnic State University, San Luis Obispo, CA 93407, USA; 5Academia de Ciencias de Cuba, La Habana 12400, Cuba

**Keywords:** virus, infection, microbiome, fucoidan, random forest, Oropuche, dengue

## Abstract

**Background/Objectives:** This study aimed to examine the effects of a Fucoidan-rich extract from *Saccharina latissima* (SLE-F) on differential gut microbiota composition, intestinal inflammation status, and microbial functional gene expression in participants infected with Dengue or Oropouche virus at the Hermanos Ameijeiras Hospital in Havana, Cuba. **Methods**: Fecal samples were collected at baseline, day 28, and day 90 from 90 healthy adults, some of whom contracted the virus during the study period. Functional gene analysis was conducted using two approaches—the Kruskal–Wallis H test and linear discriminant analysis effect size—applied to ortholog-level data normalized by read count and gene copy number. **Results**: Infected participants exhibited significantly lower *Lachnospiraceae*-to-*Enterobacteriaceae* (LE) ratios, indicating increased intestinal inflammation. High-dose SLE-F treatment led to a significant reduction in the LE ratio (*p* = 0.006), suggesting a strong anti-inflammatory effect. Microbiome analysis revealed a shift from dysbiosis to a more balanced composition by the end of the study, characterized by increased abundances of *Akkermansia muciniphila*, *Bifidobacterium adolescentis*, and *B. longum*, along with decreased pro-inflammatory taxa such as *Fusobacterium*. **Conclusions**: Genetic analysis provided distinct yet complementary insights into the microbiome’s functional responses to infection and therapeutic modulation by Fucoidan. These findings highlight the therapeutic potential of high-dose Fucoidan in reducing gut inflammation and promoting microbiome recovery following viral infections.

## 1. Introduction

The gut microbiome, a diverse and dynamic community of microorganisms within the human digestive system, plays a pivotal role in maintaining overall health [1]. It is crucial for regulating metabolism, modulating immune responses, and protecting against harmful infections [2]. Emerging evidence indicates that the gut microbiota can both inhibit and facilitate viral infections, depending on the microbial composition and host context.

In particular, the gut microbiome significantly influences host immune responses to arthropod-borne viral infections, such as dengue virus [3]. Studies have implicated specific microbial shifts, notably an overrepresentation of Pseudomonadota, in the pathogenesis of severe dengue, suggesting impaired microbial regulation and heightened inflammatory responses [4]. By modulating immune cell activation and signaling pathways, the microbiota may either enhance antiviral defenses or, when dysregulated, exacerbate disease severity [5].

Recent research supports the notion that a healthy and diverse microbiome bolsters antiviral immunity. For instance, Liang et al. (2024) demonstrated that *Cetobacterium somerae* inhibits viral infection in zebrafish via the TLR2–type I interferon axis [6]. Similarly, Chancharoenthana et al. (2022) reported associations between gut dysbiosis, altered blood bacteriomes, and progression to severe dengue [2].

In 2024, Cuba reported 626 confirmed cases of Oropouche. These cases were detected through the surveillance of nonspecific febrile syndrome (NFS), with occurrences registered in 109 municipalities across the 15 provinces of the country [7,8]. During the same period, Cuba also experienced cases of dengue virus [8,9] and Sars-CoV-2 infection (https://www.paho.org/es/documentos/alerta-epidemiologica-sars-cov-2-influenza-otros-virus-respiratorios-region-americas-5 Accessed 5 May 2025).

Fucoidans are sulfated polysaccharides primarily found in brown seaweeds and have been increasingly recognized for their diverse biological properties, including anti-inflammatory, antiviral, immunomodulatory, and prebiotic effects [10,11,12]. Among these, Saccharina latissima produces a well-characterized fucoidan with high purity, structural consistency, and favorable bioavailability profiles [13,14]. Several preclinical studies have demonstrated that fucoidans can modulate gut microbiota composition by promoting beneficial anaerobes (e.g., *Lachnospiraceae*) and suppressing pro-inflammatory taxa (e.g., *Enterobacteriaceae*) [15]. In addition, fucoidan has been shown to reduce lipopolysaccharide (LPS)-induced inflammation and may interfere with viral attachment or replication, making it a promising candidate for managing virus-associated dysbiosis [16,17,18]. These characteristics support the selection of *S. latissima*-derived fucoidan as an intervention targeting inflammation-associated microbial imbalance in this study.

In this complex epidemiological landscape, we conducted a double-blind, placebo-controlled clinical trial to evaluate the effects of a bioactive fucoidan-rich extract derived from the brown seaweed Saccharina latissima (SLE-F) on gut microbial composition and functionality. Concurrently, we characterized the baseline gut microbiome of the healthy Cuban population. This dual approach provided an opportunity to investigate both the role of the microbiome in recovery from systemic viral infections and the potential therapeutic benefits of SLE-F during acute viral illness. We hypothesized that systemic viral infections would disrupt gut microbial homeostasis, resulting in a decreased LE ratio (*Lachnospiraceae*-to-*Enterobacteriaceae* ratio)—a validated microbial biomarker of intestinal inflammation [19,20]. Lower LE ratios are associated with dysbiosis and a pro-inflammatory gut environment, while higher ratios reflect microbial balance and intestinal health. Furthermore, we posited that treatment with SLE-F would mitigate these disruptions, normalize the LE ratio, and reduce intestinal inflammation. Furthermore, we posited that treatment with SLE-F would mitigate these disruptions, normalize the LE ratio, and reduce intestinal inflammation as validated at the gene level.

## 2. Materials and Methods

### 2.1. Materials

The investigational product used in this clinical trial was a high-purity fucoidan extract (≥75% *w*/*w*), derived from the brown seaweed Saccharina latissima (SLE-F), and supplied by OCEANIUM^®^ (Oban, Scotland, UK). For the trial, SLE-F was encapsulated at a dose of 500 mg per capsule. The placebo consisted of 125 mg of microcrystalline cellulose and was carefully designed to be visually and physically indistinguishable from the active formulation. Both the active and placebo capsules were composed of clear hydroxypropyl methylcellulose (HPMC) and packaged in identical blister packs to ensure product integrity, maintain blinding, and support accurate administration throughout the study.

### 2.2. Human Safety Assessment

This trial investigated the effects of SLE-F on gut microbial composition and functionality in a three-arm, double-blind, placebo-controlled design. Ninety-one healthy adult participants were enrolled and assigned to a dose of 500 mg/capsule, twice daily, or a placebo (125 mg/capsule, twice daily). The trial spanned 90 days.

The safety assessment included monitoring for adverse events, changes in hematological and biochemical parameters, transit time, and overall health status using the validated SF-36 health questionnaire [21]. Regular clinical evaluations and sample collections were conducted at baseline (Baseline), day 30, and the end of the study.

### 2.3. Study Design

This randomized, double-blind, placebo-controlled clinical trial evaluated the effects of a single daily dose of SLE-F (500 mg) on gut microbiota composition and functionality in healthy adults. A total of 91 participants were initially recruited from the Hermanos Ameijeiras Hospital in Havana, Cuba. The study followed the principles of Good Clinical Practice and the Declaration of Helsinki [22], and received ethical approval from the Hermanos Ameijeiras Ethics Committee (CEI-HHA-04), the National Institute of Nutrition of Cuba (PI50456/24), and the Cuban Ministry of Health. The trial is registered in the Cuban Public Registry of Clinical Trials under the identifier RPCEC00000443.

### 2.4. Sample Size

Sample size calculations were based on a 95% confidence level with a 10% margin of error, targeting the hospital’s staff population of approximately 3000 individuals. Using the QuestionPro online calculator [23], a required sample size of 94 was estimated. For logistical and feasibility reasons, the final sample included 90 participants.

### 2.5. Study Population

Eligible participants were healthy adults aged 18–65 years, without restriction by gender or skin color, ensuring a demographically diverse sample. All participants provided written informed consent after receiving a full explanation of the study’s objectives, procedures, and potential risks.

Exclusion criteria included known allergies to fish or shellfish, current or recent use of probiotics, prebiotics, or anti-inflammatory medications, diagnosed dysbiosis, or regular consumption of seaweed-containing foods. Participants who declined to consent were also excluded. Withdrawal criteria included intolerable adverse effects, the onset of comorbidities, or the use of non-study probiotic or prebiotic formulations during the trial period.

Of the 91 participants initially enrolled, one individual did not receive the assigned treatment and was excluded from all analyses following the modified intention-to-treat (mITT) approach. The remaining 90 participants were randomly assigned to one of two study arms:

A total of 90 participants were included in the modified intention-to-treat (mITT) analysis population. Subjects were randomly assigned into three groups as follows:Group A: 125 mg SLE-F, twice daily (n = 29; 32.2%).Group B: 500 mg SLE-F, twice daily (n = 30; 33.3%).Group C: Placebo, twice daily (n = 31; 34.4%).

### 2.6. Sample Collection

Fecal samples were obtained from participants enrolled in the clinical trial, which was conducted in accordance with international ethical guidelines and approved by the Ethics Committee of the Hermanos Ameijeiras Clinical and Surgical Hospital. Prior to enrollment, all participants provided written informed consent.

Participants were instructed to collect stool samples using sterile kits provided by the study team. Collection was done by sampling the center of the stool and placing it on clean paper to avoid contact with water or other contaminants. Samples were immediately transferred into tubes pre-filled with DNA/RNA Shield (Zymo Research, Irvine, CA, USA), a preservation solution designed to stabilize microbial nucleic acids at room temperature. Although the preservative allows ambient storage and transport, all samples were ultimately frozen at −80 °C for long-term storage and optimal integrity.

Samples were collected at three time points: baseline (Day 0), Day 28, and end of study (Day 90). For the present microbiome analysis, only baseline and end-of-study samples were used to assess longitudinal changes within individuals. This approach helped control for natural inter-individual variability in microbiome composition.

Sample shipments were managed under tightly controlled conditions to maintain the stability of nucleic acids during transit. Once received, samples were processed and analyzed at EzBiome (Gaithersburg, MD, USA), where downstream microbiome profiling was performed.

### 2.7. 16S rRNA Gene Metagenomic Sequencing

Microbial community profiling was conducted through 16S rRNA gene sequencing by EzBiome (Gaithersburg, MD, USA). DNA concentrations were quantified using the QuantiFluor^®^ dsDNA System in conjunction with a Quantus™ Fluorometer (Promega, Madison, WI, USA). The hypervariable V3–V4 region of the 16S rRNA gene was targeted using a primer set that included Illumina adapter overhangs. The primer sequences used were as follows:Forward (IlluminaF): CCTACGGGNGGCWGCAG.Reverse (IlluminaR): GACTACHVGGGTATCTAATCC.

An initial round of PCR was performed in 25 µL reactions, each containing 12.5 ng of extracted DNA, 12.5 µL of 2× KAPA HiFi HotStart ReadyMix (Kapa Biosystems, Wilmington, MA, USA), and 5 µL of 1 µM primers. Thermal cycling conditions were as follows: denaturation at 95 °C for 3 min; 25 cycles of 95 °C for 30 s, 55 °C for 30 s, and 72 °C for 30 s; followed by a final extension at 72 °C for 5 min. PCR amplicons were purified using Mag-Bind^®^ RxnPure Plus magnetic beads (Omega Bio-tek, Norcross, GA, USA).

A secondary PCR was performed to incorporate sample-specific barcodes and sequencing adapters. This reaction followed the same master mix formulation, with thermal cycling involving 8 amplification cycles after an initial 3 min denaturation step. Products were purified and normalized using the Mag-Bind^®^ EquiPure Library Normalization Kit (Omega Bio-tek). Final libraries were pooled, assessed for quality and fragment size using an Agilent 2200 TapeStation, Santa Clara, CA USA, and sequenced on the Illumina MiSeq platform, San Diego, CA USA using 2 × 300 bp paired end reads.

### 2.8. Bioinformatic Processing and Quality Control

Raw sequencing data were processed using the QIIME 2 platform (version qiime2-amplicon-2023.9) [24], a widely used, open-source bioinformatics framework known for its modular and reproducible workflow design. Demultiplexed paired-end reads were analyzed with the DADA2 plugin [25], which performs quality filtering, denoising, and chimera removal in a unified step to generate high-resolution amplicon sequence variants (ASVs). This method corrects sequencing errors specific to Illumina platforms, allowing for accurate taxonomic resolution down to the strain level [25].

DADA2 identifies and eliminates chimeric sequences—common artifacts of PCR amplification—using a reference-free approach built from the dataset itself [25]. This ensures that biologically relevant sequences are retained while minimizing noise from spurious variants.

The resulting outputs included a feature table indicating the abundance of each ASV across samples, along with a corresponding set of representative sequences. These were used in subsequent taxonomic classification and diversity analyses. Quality control metrics such as read length distributions and per-base quality scores were assessed using QIIME 2’s built-in visualization tools, providing detailed insight into sequencing performance.

ASVs were selected over traditional operational taxonomic units (OTUs) due to their superior taxonomic resolution and ability to preserve fine-scale ecological structure without relying on arbitrary clustering thresholds [26]. This approach enhances the precision and interpretability of community composition analyses.

### 2.9. Alpha Diversity Analysis

Alpha diversity was assessed to evaluate within-sample microbial diversity across different time points and treatment groups. Two commonly used ecological indices were employed: the Shannon diversity index and the Simpson diversity index. These metrics capture complementary aspects of community structure—namely, species richness and evenness.

All diversity analyses were performed using the QIIME 2 platform (version qiime2-amplicon-2023.9) [27]. The Shannon index was calculated to quantify both the abundance and evenness of the microbial taxa present, giving greater weight to rare taxa. The Simpson index, which emphasizes dominant taxa, was also computed to assess community evenness.

Raw 16S rRNA gene sequencing data were processed via the DADA2 [25] pipeline within QIIME 2 to generate high-resolution amplicon sequence variants (ASVs). These ASV tables were then rarefied to a uniform sequencing depth to control for sampling effort and sequencing bias across samples. Alpha diversity metrics were calculated using the qiime diversity alpha plugin.

Group comparisons for diversity metrics were conducted using the Kruskal–Wallis test for non-parametric data [28]. *p*-values < 0.05 were considered statistically significant. All visualizations of alpha diversity indices (e.g., boxplots) were generated using the qiime EMPeror, v. 1.0.4 [29] and matplotlib, v. 3.10.3 [30] packages.

### 2.10. Random Forest Classifier Analysis

To identify microbial and genetic features associated with viral infection status, a supervised machine learning approach was implemented using a Random Forest (RF) classifier—a robust ensemble algorithm that constructs multiple decision trees during training and aggregates their outputs via majority voting to perform classification tasks [31,32].

Before training the model, feature data representing bacterial family-level relative abundances were standardized (z-score normalization) to ensure comparability and prevent scale bias. The dataset was randomly partitioned into training (70%) and testing (30%) subsets, with stratified sampling to preserve the original distribution of infected and uninfected samples.

Model performance was assessed using several key metrics: classification accuracy, the area under the receiver operating characteristic curve (AUC-ROC), and feature importance scores. The latter quantifies the relative contribution of each bacterial family to the model’s predictive performance, aiding in biological interpretation. To enhance model reliability and mitigate overfitting, 5-fold cross-validation was applied across the entire dataset.

This analytical strategy allowed for the identification of microbial taxa that discriminated between infected and uninfected individuals, supporting further exploration of microbial biomarkers associated with systemic viral infections.

### 2.11. Recovery Score

To evaluate the extent of gut microbiome recovery in virus-infected participants by the end of the study (EOS), a recovery score was calculated based on the Bray–Curtis dissimilarity index. This ecological distance metric quantifies differences in community composition between two groups—in this case, the taxonomic profiles of the infected EOS cohort and those of the healthy EOS cohort. Bray–Curtis dissimilarity values [33] range from 0 (indicating identical community structure) to 1 (complete dissimilarity). To facilitate interpretation, the recovery score was defined as:Recovery Score = 1 − Bray–Curtis Dissimilarity

Higher recovery scores (approaching 1) indicate greater similarity to the healthy microbiome composition, reflecting a more complete recovery of microbial structure following infection. This approach provides a quantitative means of comparing microbiome restoration between treatment groups and identifying patterns of resilience or dysbiosis resolution in response to intervention.

### 2.12. Abundance and Biomarker Analysis

To minimize false positives, an in-house script quantified reads mapping to a species only when the total coverage of its core genes (for archaea and bacteria) or genome (for fungi and viruses) exceeded 25%. Species abundance was determined based on the total number of mapped reads, and normalized species abundance was calculated by dividing by the total length of all references for the respective species.

To infer microbial community function from 16S profiles, the ASV table was processed using PICRUSt2 (Phylogenetic Investigation of Communities by Reconstruction of Unobserved States, version 2.4) [34]. PICRUSt2 maps ASVs to reference genomes using a phylogenetic placement approach and predicts gene family abundances based on evolutionary modeling. KEGG Orthology (KO) identifiers were used to annotate predicted gene functions. Copy number normalization was applied to correct for 16S rRNA gene copy variation across taxa, and results were rarefied to 1000 reads per sample to standardize sequencing depth across samples.

### 2.13. Statistical Analysis: LEfSe and Kruskal–Wallis Tests

To identify differentially abundant microbial functions, KO-level abundance tables were analyzed using two complementary statistical approaches:

#### 2.13.1. LEfSe Analysis

LEfSe (linear discriminant analysis effect size) was used to identify orthologs that were statistically different and had consistent effects between treatment groups [35,36]. The LEfSe pipeline involved three steps: (i) a non-parametric Kruskal–Wallis test to detect significant differences across groups, (ii) an optional pairwise Wilcoxon test [37] for subclass comparisons, and (iii) linear discriminant analysis (LDA) to estimate the effect size of each differentially abundant feature. Features with a *p*-value < 0.05 and an LDA effect size > 2.0 were considered significant. Group comparisons focused on infected participants receiving SLE-F treatment (Group B) versus those receiving placebo.

#### 2.13.2. Kruskal–Wallis H Test (Independent Analysis)

As an independent, non-parametric approach to detect statistically significant differences in ortholog abundances between groups, the Kruskal–Wallis H test [28] was applied to each KO. *p*-values were adjusted for multiple comparisons using the Benjamini–Hochberg False Discovery Rate (FDR) method [38]. Orthologs with FDR-adjusted *p*-values < 0.05 were retained for interpretation. Fold change calculations were performed by comparing median normalized abundances at EOS to Day 1 within each group, and cross-group differences were examined to determine directional trends.

Together, these analyses enabled the identification of both statistically significant and biologically relevant functional biomarkers associated with SLE-F intervention during post-infection microbiome recovery.

### 2.14. Statistical Analysis

All statistical analyses were conducted using R (version 4.3.0) and Python (version 3.10) to ensure a robust and reproducible workflow. Comparisons of taxonomic and functional abundances across the three study time points (baseline, 28D, and EOS) were performed using non-parametric statistical tests. Specifically, the Wilcoxon rank-sum test was applied for pairwise comparisons, and the Kruskal–Wallis test was used for group-wise comparisons. To account for multiple testing, false discovery rate (FDR) correction was applied, ensuring that significant results had a reduced likelihood of false positives.

Correlations between microbial taxa and predicted functional pathways were analyzed using Spearman’s rank correlation, which is suitable for non-parametric data and captures monotonic relationships between variables. Key microbial groups and functional pathways, such as those associated with short-chain fatty acid (SCFA) production and lipopolysaccharide (LPS) biosynthesis, were examined for trends over time. These trends were visualized using line plots to depict longitudinal changes and bar charts to highlight differences in relative abundances or pathway contributions. This comprehensive statistical approach provided detailed insights into the dynamics of the gut microbiome and its functional potential across the intervention period.

## 3. Results

While participants were randomized into three groups, only Groups B and C were included in the primary comparative analyses. This decision was based on prior internal findings indicating that microbiome and inflammation outcomes in Group A (low-dose SLE-F) were consistently similar to those observed in the placebo group and thus were excluded to improve analytical clarity and focus.

### 3.1. Epidemiological Context and Participant Infection Status

During the study, viral infections identified among participants included Dengue virus and Oropouche virus, based on national epidemiological surveillance data and clinical records.

Due to the epidemiological situation during the study where the incidence of Oropuche [39], Dengue [40,41] and COVID [42] were increased, in a little more than 50% of the subjects globally (52.2%) the presence of some viral infection was found (Table 1). No differences between the groups are detected in the presence of infection, nor is dependence between the time of infection and treatment, although it is noted that 58.8% of the subjects assigned to the Placebo group presented infection at the beginning of the beginning of the study.

### 3.2. Baseline Comparison of Intestinal Inflammation

At baseline, the *Lachnospiraceae*-to-*Enterobacteriaceae* (LE) ratio—a marker of intestinal inflammation [20]—was compared across participants with and without recent viral infections. Baseline Inflammation Assessment Using LE Ratio.

At baseline, intestinal inflammation was assessed using the *Lachnospiraceae*-to-*Enterobacteriaceae* (LE) ratio—a validated microbiome-derived biomarker of gut inflammatory status [11]. Participants were stratified into healthy controls and those with recent viral infections (Oropouche or Dengue virus), and LE ratios were compared between these cohorts to establish a pre-intervention inflammatory baseline. As shown in Figure 1, the healthy group exhibited significantly higher LE ratios than the infected group (*p* = 0.012), indicating lower baseline inflammation. This statistically significant difference underscores the inflammatory disruption associated with acute viral infection and provides a reference point for evaluating the potential therapeutic impact of Fucoidan.

### 3.3. Functional Gene Signatures: Random Forest

Random Forest modeling was employed to identify bacterial families and functional orthologs that best distinguished between virus-infected and uninfected individuals, as well as between Fucoidan-treated (SLE-F) and placebo cohorts at end of study. The model achieved an AUC-ROC score of 0.71, indicating a moderate discriminatory ability. It demonstrated stronger performance in identifying uninfected individuals than infected ones, likely due to sample size imbalance and infection heterogeneity.

Figure 2 shows the bacterial families contributing most to the classification. *Clostridiaceae*, *Enterobacteriaceae*, and *Fusobacteriaceae* emerged as the top contributors and were overrepresented in the infected group, consistent with their established roles in inflammation and dysbiosis. In contrast, families such as *Bifidobacteriaceae*, *Lachnospiraceae*, and *Ruminococcaceae* had lower importance scores, aligning with their association with homeostasis and post-infection recovery.

The model also identified several high-importance orthologs (Table 2) whose differential abundance distinguished SLE-F from placebo-treated participants. These included genes involved in chemotaxis (K08191), glycoside hydrolases (K16785), and mucosal interaction (K10118), suggesting a shift in functional microbial traits in response to Fucoidan. These functional changes mirror the taxonomic signals and further support the therapeutic potential of Fucoidan in reshaping gut microbial activity.

Random Forest analysis identified distinct microbial orthologs that differentiate Fucoidan-treated (SLE-F) participants from placebo controls at the end of study. Notably, several genes involved in chemotaxis (e.g., K08191), carbohydrate transport and metabolism (e.g., K10118, K16785), and mucosal interaction were enriched in the SLE-F group. These included genes from transport systems and glycoside hydrolases. In contrast, oxidoreductase activity (K19353) and bile acid-inducible proteins (K21903) were more abundant in the placebo group, suggesting a divergence in metabolic functionality. These results reflect underlying taxonomic differences, where commensals associated with resilience and mucosal health appear more functionally active in the Fucoidan group, while functions tied to stress and inflammation predominate in the placebo group.

The Random Forest model identified several high-importance orthologs that distinguished SLE-F-treated from placebo-treated participants. K08191 (methyl-accepting chemotaxis protein) was significantly enriched in the SLE-F group (0.039 vs. 0.016 in Placebo EOS). K06921, an uncharacterized protein, was also elevated in the SLE-F group. Other SLE-F enriched orthologs included K10118 (ABC transporter ATP-binding protein) and K16785 (glycoside hydrolase), both involved in carbohydrate metabolism. In contrast, K19353 (oxidoreductase) and K21903 (bile acid-inducible protein) were more abundant in the placebo group. These differences reflect functionally distinct microbial profiles across treatment conditions.

### 3.4. Taxonomic Shifts in the Gut Microbiome Following Viral Infection

Longitudinal analysis of gut microbiota composition in virus-infected participants revealed consistent taxonomic changes from Day 1 to the end of study (EOS).

At the phylum level, Proteobacteria (Pseudomonadota) and Fusobacteriota showed a marked decrease in relative abundance by EOS. *Bacteroidota* also decreased slightly over the study period. In contrast, Actinomycetota and Verrucomicrobiota increased in abundance. The relative abundance of Bacillota remained largely stable across time points (Figure 3).

At the genus level, *Akkermansia* and *Bifidobacterium* exhibited notable increases by EOS. Blautia also showed a modest increase. Genera associated with lower microbial diversity at baseline, including *Fusobacterium* and *Clostridium*, declined over time. Bacteroides remained one of the dominant genera but showed a slight downward trend from baseline to EOS (Figure 4).

At the species level, *A. muciniphila* increased significantly by EOS (*p* = 0.0046). Within *Bifidobacterium*, both *B. adolescentis* (*p* = 0.0005) and *B. longum* (*p* = 0.0036) showed significant increases. *Blautia wexlerae* decreased over time (*p* = 0.1025). Increases were also observed in *Faecalibacterium prausnitzii* (*p* = 0.034) and *Fusicatenibacter saccharivorans* (*p* = 0.016) (Figure 5).

### 3.5. Alpha Diversity Trends

Alpha diversity was measured using the Shannon and Simpson indices in virus-infected participants at Day 1 and End of Study (EOS). Both indices increased significantly by EOS (Shannon, *p* = 0.0248; Simpson, *p* = 0.0257), indicating higher within-sample microbial richness and evenness at the study’s conclusion (Figure 6).

### 3.6. Impact of Fucoidan Treatment on LE Ratios and Functional Gene Profiles

The *Lachnospiraceae*-to-*Enterobacteriaceae* (LE) ratio was evaluated in virus-infected participants stratified into three groups: Group A (low-dose SLE-F), Group B (high-dose SLE-F), and Group C (placebo). By end of study (EOS), Group B showed a significant reduction in LE ratio (*p* = 0.006), while Group A exhibited a non-significant decrease (*p* = 0.22). No significant change was observed in the placebo group (*p* = 0.16) (Figure 7).

PICRUSt-based functional predictions identified differential enrichment of microbial orthologs between treatment arms (Table 3). In fucoidan-treated participants, LEfSe analysis revealed increased abundance of orthologs such as K02003 (ABC transporter), K02429 (L-fucose permease), and K03773 (protein-folding isomerase). In contrast, Kruskal–Wallis analysis showed elevated levels of K15965 (glycosyltransferase) and K03124 (TFIIB) in placebo-treated individuals.

## 4. Discussion

This study explored the impact of SLE-F treatment on microbial functional gene content in Dengue and Oropouche-infected participants by comparing two analytical approaches—Kruskal–Wallis H test and Linear Discriminant Analysis Effect Size (LEfSe)—applied to ortholog-level data normalized by read count and gene copy number. The results from both methods revealed distinct but complementary insights into the microbiome’s functional responses to infection and therapeutic modulation by SLE-F regarding an integrated perspective of both, intestinal inflammation status and microbiota relative composition.

### 4.1. Baseline Comparison of Intestinal Inflammation

The observed differences in LE (*Lachnospiraceae*-to-*Enterobacteriaceae*) ratios offer important insight into the intestinal inflammatory response linked to Oropouche and Dengue virus infections. While healthy individuals generally exhibited higher LE ratios indicative of low intestinal inflammation, notable inter-individual variability was present, suggesting baseline differences in gut homeostasis. In contrast, infected participants consistently showed significantly lower LE ratios (*p* = 0.012), reflecting elevated levels of gut inflammation. These findings underscore the disruptive impact of viral infections on the gut microbiome and support the LE ratio as a potential biomarker of intestinal inflammation in the context of acute viral illness [20,43].

The LE ratio reflects the relative abundance of strict anaerobes, such as *Lachnospiraceae*—typically associated with short-chain fatty acid production and intestinal homeostasis—versus facultative anaerobes, such as *Enterobacteriaceae*, which often bloom under inflammatory conditions [32,33]. Inflammation disrupts the redox environment of the gut, favoring oxygen-tolerant taxa that are also known producers of lipopolysaccharides (LPS), a key driver of mucosal and systemic immune activation [34]. Our prior work [17] demonstrated that this ratio correlates with serum LPS levels, validating its use as a microbial surrogate of inflammation. The utility of the LE ratio [35] in this context is further supported by a growing body of literature describing microbiome compositional shifts as both indicators and contributors to inflammatory states [32,36].

The marked reduction in LE ratios among infected individuals may be linked to systemic cytokine responses triggered by viral pathogens [44]. Pro-inflammatory cytokines such as TNF-α, IL-6, and IFN-γ are known to compromise gut epithelial barrier integrity, promote bacterial translocation, and alter microbiota composition [45]. These changes can lead to the expansion of facultative anaerobes like *Enterobacteriaceae*, while reducing beneficial obligate anaerobes such as *Lachnospiraceae* [46]. However, our data suggest a more nuanced functional disruption: despite taxonomic shifts, the LEfSe and Kruskal–Wallis analyses reveal that infected individuals exhibit higher abundances of microbial orthologs associated with virulence, stress adaptation, and dysbiosis. These include K15965 (glycosyltransferase), linked to biofilm formation and immune evasion [47], and K02315 (DNA replication protein *DnaC*), reflecting uncontrolled microbial proliferation [48]. In contrast, healthy or SLE-F-treated individuals showed enrichment of functions consistent with microbial resilience and host adaptation, such as K03773 (peptidyl-prolyl isomerase), K02429 (L-fucose permease), and K02003 (ABC transporter) [49,50]. These genes are often found in beneficial commensals like *Lachnospiraceae* and are involved in protein folding, mucosal glycan metabolism, and nutrient transport [51].

Thus, the reduction in LE ratios reflects more than a taxonomic inversion; it parallels a shift in the functional capacity of the gut microbiome toward a pro-inflammatory, less regulated state. This integrated perspective emphasizes the interconnectedness between viral infection, immune response, and microbiome function, highlighting the gut as both a target and a contributor to systemic inflammation. Therapeutic interventions—such as SLE-F—that modulate microbial function may help restore intestinal homeostasis and mitigate inflammation during viral illness [35,36,37].

The observed differences in LE ratios offer important insight into the intestinal inflammatory response linked to Oropouche and Dengue virus infections. While healthy individuals generally exhibited LE ratios indicative of low intestinal inflammation, notable inter-individual variability was present, suggesting differences in baseline gut homeostasis. In contrast, infected participants consistently showed significantly lower LE ratios, reflecting elevated levels of intestinal inflammation. These findings highlight the disruptive impact of viral infections on gut health and underscore the LE ratio as a potential biomarker of inflammation in the context of acute viral illness [52].

While *Lachnospiraceae* typically produce beneficial short-chain fatty acids (SCFAs), their increased abundance in the context of viral infection may reflect a compensatory response to the inflammatory environment. Conversely, the observed reduction in *Enterobacteriaceae* might be attributed to host immune responses aimed at controlling the growth of these potentially pathogenic bacteria [53,54].

Overall, these results underline the interconnectedness between viral infections, immune responses, and gut health, highlighting the gut as a potential target for therapeutic interventions aimed at mitigating inflammation [55].

### 4.2. The Random Forest Model

The Random Forest analysis identified several orthologs that effectively distinguish between the SLE-F (Fucoidan-treated) and Placebo end-of-study (EOS) cohorts, providing insight into functionally relevant microbial adaptations associated with Fucoidan intervention. One notable feature was the enrichment of K08191, a gene encoding a methyl-accepting chemotaxis protein (MCP), in the SLE-F group. MCPs are integral to bacterial chemotaxis, enabling microbes to detect environmental gradients and migrate toward favorable conditions. The increased abundance of K08191 suggests that Fucoidan treatment may promote a more motile and environmentally responsive microbiome, potentially enhanced colonization of protective niches or improved nutrient acquisition under post-infectious stress [56,57].

Another enriched feature in the SLE-F group was K06921, an uncharacterized protein, which also displayed a higher relative abundance compared to the placebo group. Although its specific function remains unknown, its consistent elevation in the Fucoidan group implies it may play a role in microbial adaptation or host interaction within the context of mucosal recovery. Such findings are characteristic of broader trends observed across the Fucoidan-treated microbiomes: a shift toward functional traits that favor environmental sensing, resilience, and symbiosis. These orthologs, identified by their high feature importance in the Random Forest model, likely represent key microbial activities modulated by Fucoidan and underscore its potential to influence microbial recovery trajectories at a systems level [58,59,60].

An important outcome of the analysis was the identification of bacterial families that contributed most significantly to the model’s predictions. The top five families, based on feature importance, were *Clostridiaceae*, *Enterobacteriaceae*, *Bacteroidaceae*, *Fusobacteriaceae*, and *Veillonellaceae*. Among these, *Clostridiaceae* emerged as the most important contributor, possibly due to its known role in gut health and immune modulation [61]. *Enterobacteriaceae* also showed high importance, as this family includes opportunistic pathogens that may proliferate during infections [61]. Similarly, *Bacteroidaceae* is a dominant gut family with potential shifts in abundance during dysbiosis [62], while *Fusobacteriaceae* and *Veillonellaceae* are often associated with inflammatory responses and energy metabolism [63].

The families overrepresented in the infected individuals *Clostridiaceae*, *Enterobacteriaceae*, and *Fusobacteriaceae*, are often associated with inflammation, immune system activation, and gut dysbiosis, which are common responses during viral infections [63]. For instance, *Enterobacteriaceae*, which includes opportunistic pathogens, is known to proliferate under conditions of stress or immune disruption, potentially explaining its increased abundance in the infected group [63]. Similarly, *Clostridiaceae* and *Fusobacteriaceae* are families that have been linked to inflammatory conditions and shifts in gut microbiome stability, which might be triggered during viral infections [64].

The analytical results from both methods, Kruskal–Wallis H test and linear discriminant analysis effect size (LEfSe), show evident differences though complementary perceptions of the microbiome’s responsive mechanisms to viral infections and therapeutic adjustment by Fucoidan [65,66].

### 4.3. Functional Gene Signatures Differentiate Fucoidan and Placebo Cohorts

Random Forest analysis identified distinct microbial orthologs that differentiate Fucoidan-treated (SLE-F) participants from placebo controls at the end of study. Notably, several genes involved in chemotaxis (e.g., K08191), carbohydrate transport and metabolism (e.g., K10118, K16785), and mucosal interaction were enriched in the SLE-F group. These included genes from transport systems and glycoside hydrolases. In contrast, oxidoreductase activity (K19353) and bile acid-inducible proteins (K21903) were more abundant in the placebo group, suggesting a divergence in metabolic functionality. These results reflect underlying taxonomic differences, where commensals associated with resilience and mucosal health appear more functionally active in the SLE-F group, while functions tied to stress and inflammation predominate in the placebo group [67,68].

### 4.4. Kruskal–Wallis Analysis: Suppression of Dysbiosis-Associated Functions in Fucoidan-Treated Participants

The Kruskal–Wallis analysis identified several microbial orthologs with significantly different abundance profiles between groups, all of which were relatively increased in the placebo-treated infected participants but suppressed in those treated with SLE-F (Group B). Notably, suppressed genes include K09885 (aquaporin-related protein), K03124 (transcription initiation factor TFIIB), K15965 (glycosyltransferase), and K05083 (receptor tyrosine-protein kinase erbB-2 homolog). These orthologs represent functions often associated with bacterial stress adaptation, signaling, or opportunistic expansion—features commonly observed in dysbiotic microbial states following infection or antibiotic exposure [69,70].

Glycosyltransferases, for instance, are involved in cell surface modifications and may contribute to biofilm formation, immune evasion, and pathogenicity in opportunistic microbes [71,72]. Similarly, the enrichment of eukaryotic-like signaling and transcription factors (e.g., erbB-2 homologs and TFIIB) in placebo-treated subjects suggests activation of non-core microbial responses, possibly due to ecological imbalance or inflammation-driven shifts. The absence or depletion of these orthologs in Fucoidan-treated participants supports the interpretation that Fucoidan may help suppress or prevent the rise of dysbiosis-associated microbial functions, thereby facilitating a more regulated recovery process [73,74].

### 4.5. LEfSe Analysis: Promotion of Functional Resilience in the Fucoidan Group

Complementing the Kruskal–Wallis results, the LEfSe analysis highlighted a different set of orthologs that were significantly enriched in the SLE-F-treated group. These included K02003 (ABC transport ATP-binding protein), K02429 (L-fucose permease), and K03773 (FKBP-type peptidyl-prolyl isomerase). These genes reflect microbial functions associated with nutrient acquisition, protein folding, and host-microbe interaction [75,76].

ABC transporters are essential for microbial metabolic flexibility and survival in competitive or inflamed environments [77], while L-fucose permeases suggest microbial ability to utilize host-derived glycans, a trait linked with commensalism and mucosal adaptation [78]. The upregulation of protein-folding chaperones (e.g., peptidyl-prolyl isomerases) may reflect microbial efforts to maintain proteostasis under stress or during recovery, potentially supporting community stability [79,80].

These findings indicate that SLE-F may not only prevent the rise of harmful or opportunistic taxa, but also selectively enhance functions that support microbial homeostasis, mucosal symbiosis, and resilience.

#### Expanded Pathway-Level Insights from High-Dose Fucoidan EOS Samples

Additional insights emerged from pathway-level LEfSe analysis, particularly in the High-Dose SLE-f EOS cohort, which showed enriched functional categories associated with microbial recovery and mucosal interaction (Table 2). A major finding was the enrichment of ATP-Binding Cassette (ABC) transport systems, particularly orthologs K06147, K02004, and K02003. These transporters are vital for nutrient uptake and detoxification, suggesting that high-dose SLE-F promotes metabolic adaptability. Also enriched were raffinose/stachyose/melibiose transport systems (K10117, K10118, K10119), indicating enhanced capacity to utilize complex carbohydrates—a hallmark of mucosa-adapted commensals [81,82,83].

Stress response and DNA repair pathways were also significantly represented in the high-dose SLE-F EOS group, including DNA helicase II (K03657) and the chromosome partitioning protein (K03497), which may support bacterial stability under host-driven inflammatory pressures [84,85]. Furthermore, carbohydrate metabolism pathways such as glycogen phosphorylase (K00688) and isoamylase (K01214) were enriched, underscoring enhanced microbial energy cycling [86,87]. Notably, Sortase A (K07284), an enzyme involved in anchoring surface proteins to bacterial cell walls, was also elevated, suggesting increased microbial engagement with the mucosal environment [88].

### 4.6. Functional Interpretation of Gene-Level Shifts in the Context of Taxonomy and Function

The functional gene profiles derived from LEfSe, Kruskal–Wallis, and Random Forest analyses provide a mechanistic framework for understanding microbiome recovery following viral infection, particularly in the context of Fucoidan treatment. Several orthologs enriched in the SLE-F group—including K08191 (methyl-accepting chemotaxis protein), K10118 (ABC transporter for oligosaccharides), and K02429 (L-fucose permease)—highlight a gene-level shift toward microbial functions associated with mucosal sensing, nutrient utilization, and host interaction. These functional enrichments parallel increases in taxa such as *Akkermansia* and *Bifidobacterium*, both known for their roles in gut barrier integrity, immune modulation, and mucin degradation [89,90,91,92].

This integrated taxonomic-functional correlation suggests that Fucoidan facilitates not just compositional shifts but also functional reprogramming of the microbiome toward a state of increased resilience and host compatibility. For example, the rise in Verrucomicrobiota and Actinomycetota—phyla enriched in *Akkermansia* and *Bifidobacterium*, respectively—coincided with functional enrichments in genes associated with mucosal carbohydrate metabolism, SCFA production, and protein homeostasis (e.g., K00688, K03773). These pathways are critical for maintaining intestinal homeostasis during and after inflammatory insult [93,94].

Conversely, orthologs such as K15965 (glycosyltransferase), K02315 (*DnaC* replication initiator), and K05083 (receptor tyrosine kinase homolog) [95], enriched in the placebo group, correspond with increased prevalence of Pseudomonadota and Fusobacteriota—taxa often associated with inflammation, pathogenicity, and dysbiosis [67,96,97]. This alignment between pro-inflammatory taxa and stress-related microbial genes underscores the dysregulated microbial state in the absence of Fucoidan intervention.

Taken together, these findings reveal that the taxonomic patterns observed in the longitudinal data are functionally validated at the gene level. Fucoidan appears to promote microbial pathways that favor mucosal colonization, epithelial resilience, and metabolic flexibility, while suppressing genes and taxa linked to inflammation and opportunistic expansion. This dual action at both taxonomic and functional levels highlights the therapeutic potential of Fucoidan as a microbiome-stabilizing agent during post-infectious recovery [98,99,100].

The functional gene profiles derived from LEfSe, Kruskal–Wallis, and Random Forest analyses provide a mechanistic backdrop for the taxonomic shifts observed during microbiome recovery. Notably, the enrichment of orthologs related to chemotaxis (e.g., K08191), mucosal carbohydrate metabolism (e.g., K10118, K02429), and protein homeostasis (e.g., K03773) in the SLE-F (Fucoidan-treated) group aligns with the observed increase in commensal taxa such as Bifidobacterium and Akkermansia. These genera are known to harbor genes supporting mucin degradation, immune modulation, and epithelial integrity, functions mirrored by the gene-level enrichments observed [101].

Conversely, orthologs associated with virulence, stress response, and uncontrolled replication—such as K15965 (glycosyltransferase) and K02315 (DnaC)—were enriched in the placebo group, consistent with the increased abundance of pro-inflammatory taxa such as *Enterobacteriaceae* and *Fusobacterium*. The Kruskal–Wallis analysis particularly highlighted this pattern, showing elevated functional signatures related to pathogenic adaptation in the absence of Fucoidan.

At the phylum level, the recovery-associated rise in Actinomycetota and Verrucomicrobiota, and the decline in Pseudomonadota and Fusobacteriota, parallels the enrichment of genes related to host-glycan metabolism, energy cycling, and microbial community regulation. This suggests that Fucoidan fosters a functional environment conducive to beneficial phyla, while suppressing genetic traits linked to dysbiosis. The stability of Bacillota, known for SCFA production, is consistent with preservation of core metabolic functions, further supported by the consistent detection of orthologs involved in carbohydrate and lipid metabolism.

Taken together, these data demonstrate that the functional ortholog shifts observed via LEfSe, Kruskal–Wallis, and Random Forest not only reinforce but also help explain the taxonomic dynamics seen at family, genus, and species levels. Fucoidan appears to promote a gene-functional environment that facilitates recovery by supporting commensal colonization, regulating host interaction pathways, and repressing dysbiosis-linked microbial behaviors [73,74,102].

### 4.7. Integrative Insight and Therapeutic Implications

The contrast between Kruskal–Wallis and LEfSe results is informative. While Kruskal–Wallis flagged functions elevated in placebo-treated individuals (i.e., traits potentially harmful or dysbiotic), LEfSe pinpointed those more prevalent in the Fucoidan group (i.e., traits potentially beneficial). This dual pattern supports the hypothesis that Fucoidan has bidirectional effects on microbiome recovery: suppressing detrimental functional signatures while promoting adaptive, host-compatible traits.

These findings align with prior research suggesting Fucoidan exerts anti-inflammatory, prebiotic, and immunomodulatory properties, potentially by shaping gut microbial composition and metabolism [73,74]. Given its ability to modulate microbiome function at a gene level, Fucoidan may serve as a promising adjunct for managing post-infectious dysbiosis or enhancing microbiome resilience in immunologically vulnerable settings.

### 4.8. Diversity

The analysis of diversity collectively suggest that the gut microbiome of the infected cohort transitioned toward a healthier, more diverse, and balanced state by the end of the study. The observed increases in diversity indices may reflect the resolution of infection-associated disruptions and the recovery of beneficial microbial populations.

These diversity metrics align with the taxonomic shifts observed at the genus and phylum levels, where beneficial taxa such as *Akkermansia* and *Bifidobacterium* increased while inflammation-associated taxa like *Fusobacterium* decreased [103,104]. These results further highlight the importance of microbial diversity as a marker of gut health and recovery.

### 4.9. Recovery Explanation

The calculated recovery score of 0.57 reflects a partial but incomplete restoration of gut microbiome composition in the untreated viral cohort by the end of the study. While certain taxa showed trends consistent with recovery, the microbial profiles of virus-infected individuals remained notably distinct from those of healthy controls. Specifically, the overrepresentation of taxa such as Actinomycetota may represent a compensatory ecological adaptation to viral disruption. Conversely, the underrepresentation of beneficial groups, including *Prevotellaceae* and *Prevotella*, underscores a lag in the re-establishment of a fully balanced microbial community [105,106,107].

The microbiome’s natural resilience and plasticity also play a critical role. Following disruptions, such as viral infections, the gut microbiome has an innate ability to rebound, aided by interactions between surviving native microbes and external factors like diet. The resolution of inflammation during the recovery phase could have further fostered an environment conducive to the growth of beneficial microbes. Species like *A. muciniphila* and *F. prausnitzii*, which are sensitive to inflammatory conditions, likely benefited from reduced gut inflammation as the immune system restored homeostasis [106].

Nevertheless, in the treated group with the high dose of SLE-F the recovery occurred with significant differences at the end of the study compared to the rest of individuals. This founding strongly suggests the potential use of this nutritional supplement during virus infection [108].

#### Microbial Activation Pathway

These findings provide insights into how specific microbial pathways may contribute to improve the recovery during dengue or Oropuche virus infection.

ABC transporters are a family of integral membrane proteins present in all cells of all species of Archaea, Eubacteria and Eukaryota. These systems are crucial for nutrient uptake, toxin extrusion, and maintaining gut epithelial integrity [109]. Their enhanced abundance in the High-Dose EOS cohort suggests increased microbial capacity to facilitate nutrient acquisition and metabolite exchange, supporting host health [110,111].

The enrichment of raffinose/stachyose/melibiose transport systems (K10117, K10118, K10119) reflects an enhanced ability of the microbiota to metabolize complex oligosaccharides. This enrichment promotes the growth of beneficial bacteria, contributing to increased microbial diversity and short-chain fatty acid (SCFA) production [112,113,114].

Pathways such as DNA helicase II (K03657) and chromosome partitioning protein (K03497) support microbial resilience under gut stressors like low pH and bile salts [115]. This stability may enhance microbial colonization and persistence in the gut environment. Additionally, the type IV secretion system protein VirD4 (K03205) [116], enriched in the high-dose EOS group, plays a role in bacterial communication and adaptability, potentially enhancing symbiotic interactions with the host. Additionally, the type IV secretion system protein VirD4 (K03205), enriched in the High-Dose EOS group, plays a role in bacterial communication and adaptability, potentially enhancing symbiotic interactions with the host [117].

The enzymes glycogen phosphorylase (K00688) and isoamylase (K01214), facilitate energy storage and carbohydrate metabolism, suggesting an active microbiota capable of utilizing glycogen and dietary fibers to sustain gut functionality [117,118].

Sortase A (K07284) is essential for anchoring surface proteins to Gram-positive bacteria, improving colonization and interaction with host immune cells. This pathway plays a key role in maintaining the gut epithelial [119].

Severe dengue virus infection can lead to hemorrhage and shock, occurring at a stage when viremia is declining and fever has subsided. The factors contributing to the sudden deterioration of the patient’s condition remain a subject of debate. Gastrointestinal symptoms, including vomiting, diarrhea, and abdominal pain, are common in severe cases. The observed correlation between serum lipopolysaccharide levels and disease severity suggests that the gut barrier is compromised [120]. The pathways identified in the High-Dose EOS cohort reflect substantial functional adaptations in the gut microbiome. Enhanced nutrient transport systems, including ABC transporters and raffinose/stachyose transport systems, suggest improved nutrient utilization and metabolic activity. DNA repair pathways and stress-response genes indicate increased microbial stability and survival under gut conditions. Enrichment of carbohydrate-degrading enzymes highlights microbial capacity to metabolize dietary fibers, promoting beneficial metabolites like SCFAs. Finally, Sortase A and related pathways underscore improved bacterial colonization and interaction with host immune systems. These findings align with prior studies that link diverse and functional microbiota to improved gut health and host outcomes during dengue viral infection [121,122].

Further research is warranted to identify factors that influence the pace and completeness of microbiome restoration, such as dietary interventions, probiotic supplementation, or host immune responses. Understanding these dynamics can inform strategies to enhance recovery and promote long-term gut health in virus infected individuals.

### 4.10. Limitations

Despite the encouraging findings, this study has several limitations that should be considered when interpreting the results:**Subset Analysis**: The observation regarding microbiome recovery post-viral infection was derived from a subset of participants who became infected during the trial. This subset was not pre-stratified or powered specifically to assess the effects of viral infection on microbiome dynamics, limiting the generalizability of the findings.**Sample Size and Statistical Power**: The number of participants infected with Dengue or Oropouche viruses within each treatment group was relatively small. This limited the statistical power for subgroup comparisons and may have contributed to the lack of significance in some outcomes, particularly in diversity measures and certain taxa-level analyses.**Timing and Heterogeneity of Infection**: Participants contracted viral infections at different time points during the study, which introduces heterogeneity in terms of exposure duration and immune response. Additionally, infections were not experimentally induced or uniformly documented via molecular diagnostics, which could influence the precision of infection status classification.**Confounding by Other Variables:** Although dietary and medication use exclusions were applied, other unmeasured factors—such as variations in baseline diet, host genetics, or environmental exposures—may have influenced microbiome composition and recovery trajectories.**Short-Term Follow-Up**: The study duration of 90 days may not have been sufficient to capture long-term microbiome stabilization or delayed effects of viral infection or SLE-F treatment. Longer follow-up would help clarify whether observed improvements persist over time.**Taxonomic Resolution**: While 16S rRNA sequencing offers valuable insights into bacterial composition, it lacks the resolution of metagenomic or metatranscriptomic approaches, limiting functional inference and the ability to capture strain-level variation.**Lack of Virome and Mycobiome Profiling**: The study focused exclusively on bacterial communities and did not assess viral or fungal components of the microbiome, which may also play critical roles in gut health and immune regulation during infection.

## 5. Conclusions

This study shows that SLE-F (Fucoidan) treatment modulates gut microbiome function in virus-infected individuals, facilitating recovery from infection-associated dysbiosis. Through integrated analysis of microbial gene content using Kruskal–Wallis, LEfSe, and Random Forest models, we observed a functional reprogramming of the gut microbiome in the Fucoidan-treated group, marked by increased expression of genes associated with mucosal interaction, metabolic flexibility, and stress resilience.

Notably, SLE-F treatment promoted the enrichment of microbial orthologs involved in nutrient transport (e.g., ABC transporters), host-glycan metabolism (e.g., L-fucose permease), and protein homeostasis (e.g., peptidyl-prolyl isomerase), while suppressing genes linked to dysbiosis and pathogenicity (e.g., glycosyltransferases, *DnaC* replication initiator). These shifts aligned with taxonomic transitions favoring beneficial commensals such as *Akkermansia* and *Bifidobacterium*, and a decline in inflammation-associated taxa like *Enterobacteriaceae* and *Fusobacteriaceae*.

Furthermore, increased microbial diversity and improved recovery scores in the high-dose SLE-F group underscore the compound’s capacity to restore gut ecological balance post-infection. Functional signatures suggest that SLE-F supports microbial pathways that enhance colonization, epithelial resilience, and immunological tolerance, which are critical for post-viral recovery.

Taken together, these findings support the hypothesis that SLE-F exerts dose-dependent therapeutic effects by modulating the microbiome at the gene level—suppressing inflammatory and dysbiotic functions while enhancing traits linked to microbial symbiosis and host health. Fucoidan may therefore represent a promising co-adjuvant intervention for supporting microbiome recovery following viral infections.

## Figures and Tables

**Figure 1 genes-16-00740-f001:**
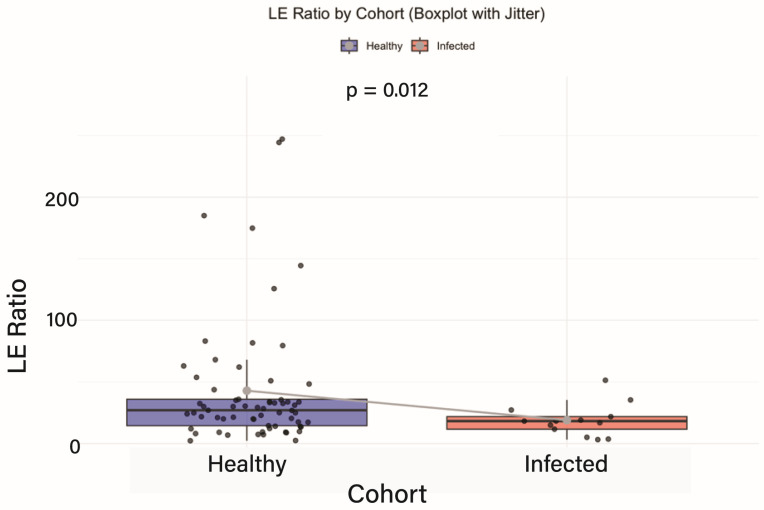
LE Ratio by cohort (boxplot with jitter). The LE ratio was significantly lower in virus-infected participants compared to healthy controls, consistent with elevated intestinal inflammation. Boxplots display interquartile ranges (IQR), with individual data points and statistical significance indicated (*p* = 0.012).

**Figure 2 genes-16-00740-f002:**
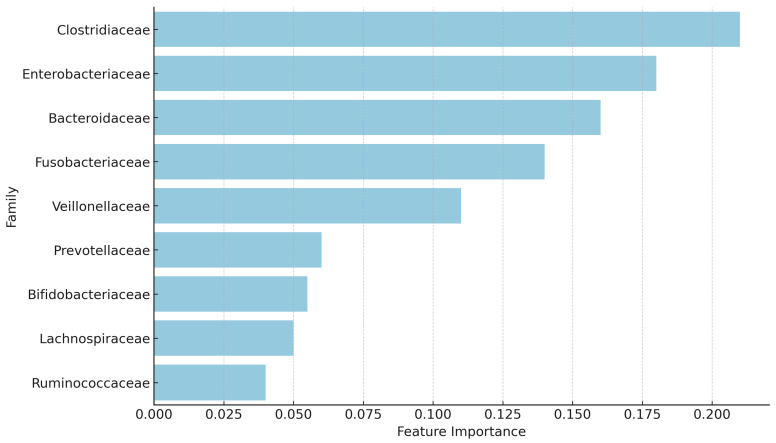
Feature importance plot from Random Forest analysis of bacterial families distinguishing infected from uninfected individuals. *Clostridiaceae*, *Enterobacteriaceae*, and *Fusobacteriaceae* showed high importance, consistent with pro-inflammatory microbial profiles.

**Figure 3 genes-16-00740-f003:**
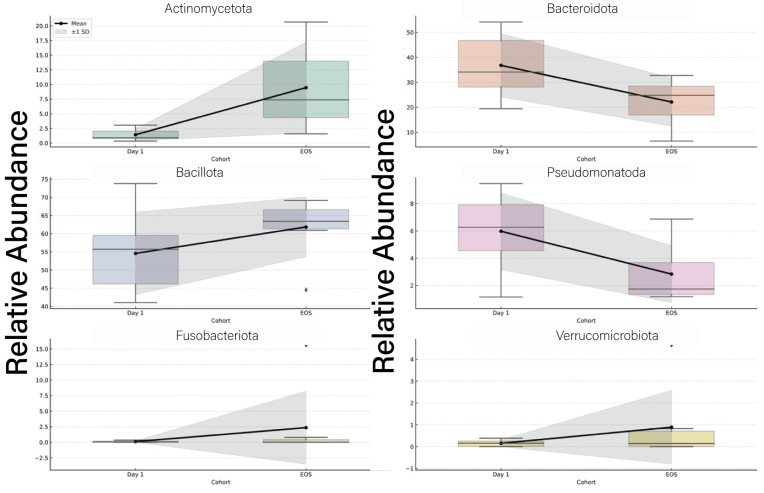
Relative abundance of major bacterial phyla in virus-infected participants at Day 1 and End of Study (EOS). Boxplots display mean values with standard deviation shading. The black line connects the group means across timepoints, indicating the trend in mean relative abundance. The black rhombus represents the group mean at each timepoint. Shaded gray areas denote ±1 standard deviation from the mean, providing a visual estimate of variability within each group. Notable trends include a decrease in Pseudomonadota, Fusobacteriota, and Bacteroidota, and an increase in Actinomycetota and Verrucomicrobiota. Bacillota remained relatively stable across the study period. These taxonomic shifts reflect changes in gut microbiome composition following viral infection.

**Figure 4 genes-16-00740-f004:**
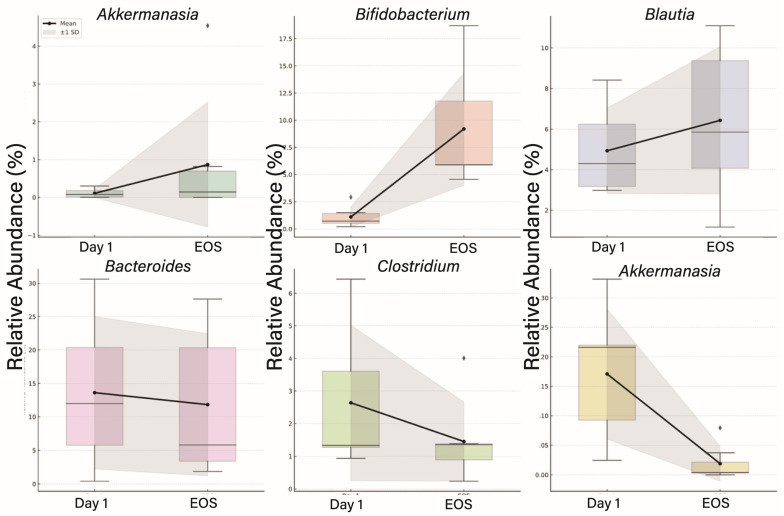
Genus-level changes in relative abundance in virus-infected participants from Day 1 to End of Study (EOS). Boxplots display mean values with standard deviation. The black line connects the group means across timepoints, indicating the trend in mean relative abundance. The black rhombus represents the group mean at each timepoint. Shaded gray areas denote ±1 standard deviation from the mean, providing a visual estimate of variability within each group. Increases were observed in *Akkermansia*, *Bifidobacterium*, and *Blautia*, while decreases were noted in *Clostridium* and *Fusobacterium*. *Bacteroides* showed a slight decline over the study period. These results indicate temporal shifts in key gut genera following viral infection.

**Figure 5 genes-16-00740-f005:**
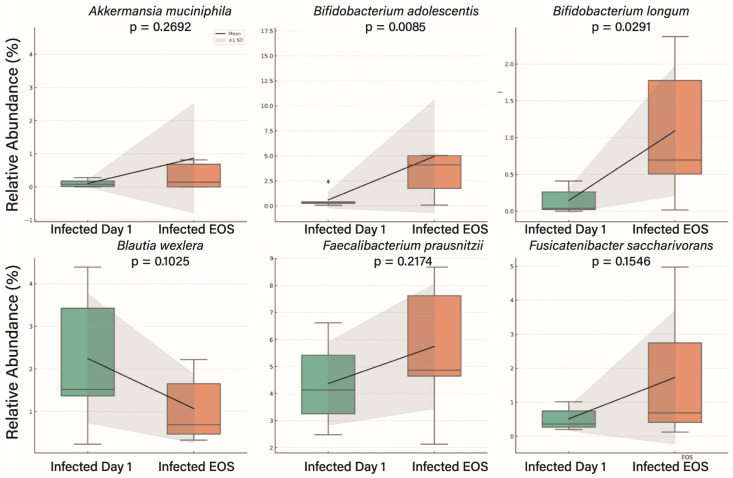
Species-level changes in gut microbiota composition from Day 1 to End of Study (EOS) in virus-infected participants. Boxplots represent relative abundance (%) with mean values and ±1 standard deviation shading. The black line connects the group means across timepoints, indicating the trend in mean relative abundance. The black rhombus represents the group mean at each timepoint. Shaded gray areas denote ±1 standard deviation from the mean, providing a visual estimate of variability within each group. Statistically significant increases were observed in *B. adolescentis* (*p* = 0.0085) and *B. longum* (*p* = 0.0291). *A. muciniphila* showed an upward trend (*p* = 0.2692). *F. prausnitzii* (*p* = 0.2174) and *F. saccharivorans* (*p* = 0.1546) also increased over time, though not significantly. Conversely (*p* = 0.1025) showed a downward trend. These results highlight species-level shifts in microbial composition during the study period.

**Figure 6 genes-16-00740-f006:**
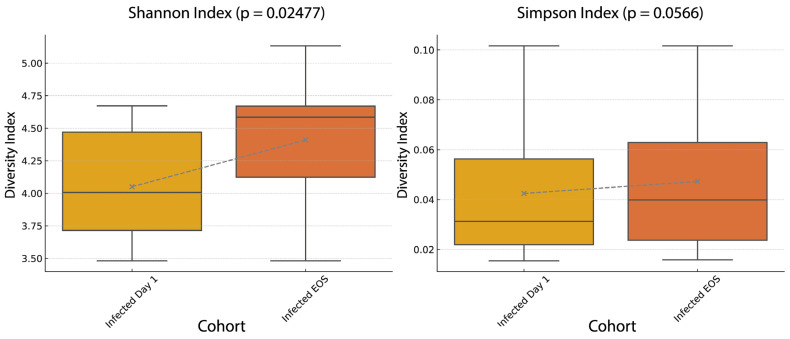
Microbial alpha diversity in infected participants at baseline and end of study. Boxplots showing changes in microbial alpha diversity in participants with confirmed viral infection, comparing samples collected at Day 1 (baseline) and end of study (EOS). The dashed line connects the group means across timepoints. The Shannon Index (**left**) shows a statistically significant increase in diversity over time (*p* = 0.02477), while the Simpson Index (**right**) shows a similar upward trend that did not reach statistical significance (*p* = 0.0566). These results suggest partial microbiome recovery following infection. Dots represent individual means, with whiskers indicating the full range and boxes the interquartile range.

**Figure 7 genes-16-00740-f007:**
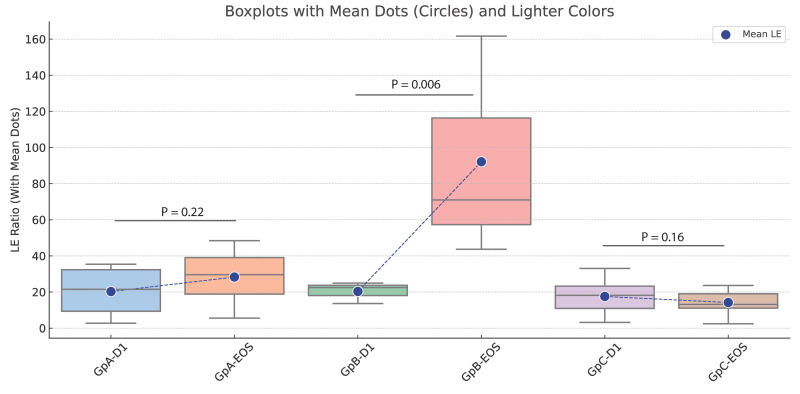
Effects of SLE-F (fucoidan) treatment on the *Lachnospiraceae*-to-*Enterobacteriaceae* (LE) ratio and predicted microbial gene functions. Boxplots display LE ratio changes across treatment groups: Group A (low-dose SLE-F), Group B (high-dose SLE-F), and Group C (placebo). A significant reduction in LE ratio was observed in Group B (*p* = 0.006). Functional predictions using PICRUSt and LEfSe identified enrichment of beneficial orthologs (e.g., K02003, K02429, K03773) in fucoidan-treated groups, while Kruskal–Wallis analysis showed elevated levels of dysbiosis-associated orthologs (e.g., K15965, K03124) in the placebo group.

**Table 1 genes-16-00740-t001:** Viral infection during study.

		Treatment			*p*-Value
		A	B	C	(χ^2^)
		29	30	31	
Viral infection during the study	No	15 (51.7%)	14 (46.7%)	14 (45.2%)	0.869
	yes	14 (48.3%)	16 (53.3%)	17 (52.2%)	
	↓	↓	↓	↓	
	Before/Beginning	4 (28.6%)	6 (37.5%)	10 (58.8%)	0.207
	During	10 (71.4%)	10 (62.5%)	7 (41.2%)	

**Table 2 genes-16-00740-t002:** Top microbial orthologs identified by random forest analysis, distinguishing SLE-F (Fucoidan) and Placebo end-of-study samples.

Ortholog	Definition	Importance Score	SLE-F EOS	Placebo EOS	Enriched Group
K01186	Sialidase	0.0588	0.0560	0.0615	Placebo
K10118	raffinose/stachyose/melibiose transport system permease protein	0.0588	0.0516	0.0455	SLE-F
K08191	MFS transporter, ACS family, hexuronate	0.0588	0.0392	0.0156	SLE-F
K06921	uncharacterized protein	0.0588	0.1570	0.0734	SLE-F
K16785	energy-coupling factor transport system permease protein	0.0471	0.0535	0.0696	Placebo
K19353	heptose-I-phosphate ethanolaminephosphotransferase	0.0431	0.0208	0.0076	SLE-F
K21903	ArsR family transcriptional regulator, lead/cadmium/zinc/bismuth-responsive transcriptional repressor	0.0353	0.0279	0.0414	Placebo
K12988	alpha-1,3-rhamnosyltransferase	0.0353	0.0378	0.0177	SLE-F
K01462	peptide deformylase	0.0353	0.0756	0.0732	SLE-F
K01607	carboxymuconolactone	0.0353	0.0452	0.0687	Placebo

**Table 3 genes-16-00740-t003:** Pathway-level functional orthologs enriched in the high-dose Fucoidan end-of-study (EOS) cohort.

**A. Kruskal–Wallis**					
**Ortholog**	**Definition**	**Fold Change SLE-F**	**Fold Change** **Placebo**	**FDR** **Adjusted *p*-Value**	**Cohort** **Enriched**
K09885	Aquaporin rerated protein, other eukaryotes	0.0180	1.0000	0.0002	Placebo
K03124	Transcription initiation factor TFIIB	0.0088	1.0000	0.0002	Placebo
K15965	glycosyltransferase	0.0093	1.0000	0.0002	Placebo
K05083	Receptor tyrosine-protein kinase erbB-2	0.0180	0.1250	0.0028	Placebo
K03334	L-amino-acid oxidase	0.0076	0.1272	0.0028	Placebo
K21148	[CysO sulfur-carrier protein]-thiocarboxylate-dependent cysteine synthase	0.0053	0.1272	0.0028	Placebo
K15761	Toluene monooxygenase system protein B	0.0138	0.1250	0.0028	Placebo
K03773	Protein-folding Isomerase	0.0048	0.1272	0.0028	Placebo
K02429	L-fucose permease	0.0066	0.1272	0.0028	Placebo
K07190	Phosphorylase kinase alpha/beta subunit	0.0059	0.0673	0.0031	Placebo
**B. LEfSe (Linear Discriminant Analysis Effect Size)**					
**Ortholog**	**Definition**	**Fold Change SLE-F**	**Fold Change Placebo**	**LDA Effect Size**	**Cohort Enriched**
K07284	Sortase A	1.2755	0.6212	2.6276	SLE-F
K02315	DNA replication protein DnaC	1.5720	1.6683	2.4694	Placebo
K03773	FKBP-type peptidyl-prolyl cis-trans isomerase FklB	1.0909	0.5163	2.4286	SLE-F
K02003, K02004, K06147	ABC transport system ATP-binding protein	1.3726	1.1534	2.4191	SLE-F
K02429	MFS transporter, FHS family, L-fucose permease	1.0626	0.5566	2.3667	SLE-F
K02030	Polar amino acid transport system substrate-binding protein	1.2128	1.3541	2.2981	Placebo
K03286	OmpA-OmpF porin, OOP family	1.0185	0.5629	2.2478	SLE-F
K10117,K10118, K10119	Raffinose/stachyose/melibiose transport system substrate-binding protein	1.6093	1.4149	2.2456	SLE-F
K03657	Stress response/DNA helicase I/ATP-dependent DNA helicase PcrA	1.2597	1.3317	2.1916	SLE-F
K16785	Energy-coupling factor transport system permease protein	1.3868	1.8712	2.1724	Placebo
KO1214	Isoamylase	1.7511	1.1182	2.6544	SLE-F
KO0688	Glycogen phosphorylase	1.3336	1.1124	2.4281	SLE-F

## Data Availability

The datasets generated and analyzed during the current study are not publicly available due to participant privacy considerations but are available from the corresponding author on reasonable request.

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
