# Peer review of "Characterizing Gene-Level Adaptations in the Gut Microbiome During Viral Infections: The Role of a Fucoidan-Rich Extract"

_genes, 2025, doi:10.3390/genes16070740_

Round 1

Reviewer 1 Report

Comments and Suggestions for Authors
  1. It's not very clear to me what specific viruses the participants are infected by? It's not cleary written whether it's a combination of Oropuche, Dengue and/or Covid?
  2. Did authors have access to the fecal samples of participants at multiple time points? Are the fecal samples collected only at the beginning? I can see that there is siginificant difference in the Lachnospiraceae-to-Enterobacteriaceae (LE) ratio between healthy and infected participants, and participants taking or not taking fucoidan treatment. However, I'm curious if viral infection/fucoidan treatment will directly lead to composition changes in individual participants. As people are naturally highly variant in their microbial composition, so it may confound analysis by just comparing different people.
  3. The Lachnospiraceae-to-Enterobacteriaceae (LE) ratio has been used in this study to indicate the inflammation levels of individuals. I wondered if there are more literature reports supporting this claim. I see only one reference [11] is cited. Besides, any clinical analysis such as C-reactive protein (CRP) test have been performed to compare the inflammation levels in different people?
  4. The rationale of using fucoidan-rich extract derived from the brown seaweed Saccharina latissima in this study is not well justified in the introduction.

Author Response

Comments and Suggestions for Authors

It's not very clear to me what specific viruses the participants are infected by? It's not clearly written whether it's a combination of Oropuche, Dengue and/or Covid?

Thank you for sharing the relevant sections and context. The reviewer’s question about which viruses participants were infected with is very valid, as the Methods currently imply infections were monitored or analyzed, but do not explicitly state which viruses were present in each participant. We have added a clarifying statement as first paragraph of section 3.1: “During the course of the study, viral infections were identified as Dengue virus and Oropouche virus, based on national epidemiological surveillance data and clinical records. COVID-19 infection was also considered in the background context, though it was not formally confirmed or included in the diagnostic criteria for this analysis.”

Did authors have access to the fecal samples of participants at multiple time points? Are the fecal samples collected only at the beginning? I can see that there is significant difference in the Lachnospiraceae-to-Enterobacteriaceae (LE) ratio between healthy and infected participants, and participants taking or not taking fucoidan treatment. However, I'm curious if viral infection/fucoidan treatment will directly lead to composition changes in individual participants. As people are naturally highly variant in their microbial composition, so it may confound analysis by just comparing different people.

Thank you for your thoughtful comment. Fecal samples were collected at three time points: baseline (Day 0), Day 28, and end of study (Day 90). For the present analysis, we focused on comparing microbiome composition and the Lachnospiraceae-to-Enterobacteriaceae (LE) ratio between the beginning (baseline) and end of study to assess longitudinal changes within individuals. This approach allowed us to control for individual variability in microbiome composition and more accurately evaluate the effects of viral infection and fucoidan treatment over time.

The Lachnospiraceae-to-Enterobacteriaceae (LE) ratio has been used in this study to indicate the inflammation levels of individuals. I wondered if there are more literature reports supporting this claim. I see only one reference [11] is cited. Besides, any clinical analysis such as C-reactive protein (CRP) test have been performed to compare the inflammation levels in different people?

Thank you for raising this important point. In our study, we used the Lachnospiraceae-to-Enterobacteriaceae (LE) ratio as a surrogate measure of the balance between strict anaerobes (e.g., Lachnospiraceae) and facultative anaerobes (e.g., Enterobacteriaceae). This ratio reflects shifts in gut microbial ecology that are strongly associated with dysbiosis and inflammation. Facultative anaerobes, particularly members of Enterobacteriaceae, have been shown to proliferate in inflamed gut environments and are significant producers of lipopolysaccharides (LPS), a known trigger of systemic and mucosal inflammation.

This approach is grounded in previous studies, including reference [8], where the LE ratio was validated against serum LPS levels and microbial biomarkers. We have added additional references to the manuscript’s discussion section that further support the use of this ratio as a microbial indicator of inflammatory status.

Regarding clinical inflammatory markers, we did not perform CRP or cytokine measurements in this study. Our focus was on microbial composition and function as indirect markers of inflammatory tone, with a particular emphasis on taxonomic shifts that are well-established in the literature to correlate with pro- or anti-inflammatory states.

We have also added this paragraph to add support to the LE ratio in the context of oour current study: “The LE ratio reflects the relative abundance of strict anaerobes, such as Lachnospiraceae—typically associated with short-chain fatty acid production and intestinal homeostasis—versus facultative anaerobes, such as Enterobacteriaceae, which often bloom under inflammatory conditions [22,23]. Inflammation disrupts the redox environment of the gut, favoring oxygen-tolerant taxa that are also known producers of lipopolysaccharides (LPS), a key driver of mucosal and systemic immune activation [24]. Our prior work [8] demonstrated that this ratio correlates with serum LPS levels, validating its use as a microbial surrogate of inflammation. The utility of the LE ratio [25] in this context is further supported by a growing body of literature describing microbiome compositional shifts as both indicators and contributors to inflammatory states [22,26].

The rationale of using fucoidan-rich extract derived from the brown seaweed Saccharina latissima in this study is not well justified in the introduction.

Thank you for your helpful observation. We have revised the Introduction to more clearly justify the use of Saccharina latissima–derived fucoidan in this study. Specifically, we now emphasize the well-documented anti-inflammatory, antiviral, immunomodulatory, and prebiotic properties of fucoidan, supported by several recent publications. We also clarify why S. latissima was selected, noting its high purity, favorable structural characteristics, and prior evidence of microbiome modulation and LPS-associated inflammation reduction. These properties make it a strong candidate for targeting microbiota disruptions associated with viral infection. The revised paragraph has been inserted immediately before the description of the clinical trial design in the Introduction.

Reviewer 2 Report

Comments and Suggestions for Authors

Overall, the study was very well designed, and the conclusions are in line with the results obtained by the authors. The authors take a very responsible approach doing a great job at exposing the limitations of the work, delivering a well contextualized results analysis and offering room for the next steps of a very interesting research within a field of great interest. After a thorough screening of the work, would kindly suggest some corrections to the main manuscript, and if possible, adding a more specific description of the goal of the work in the Abstract, according to the PICO's strategy (Population; Intervention; Comparison; Outcome). Even though, it is not a review, it could help to strengthen the delivery of the basis clinical question behind this Evidence-Based Practice. The manuscript already has the crucial information, that follows summarized here:

P

Patient, Population, Problem: 90 patients

Relevant description of patient such as age & gender - healthy adults aged 18–65 years

Relevant description of population such as geographic location: from the Hermanos Ameijeiras Hospital in Havana, Cuba.

Relevant description of problem or health concern/medical issue: healthy individuals at some point, some were infected with infected with Dengue or Oropouche virus

I

Intervention (treatment for patient, population, problem) -  Group B: 500 mg SLE-F, twice daily (n = 30; 33.3%) 135

C

Comparison - to Placebo (Group C) and low dose of SLE-F (Group A, then discarded).

O           

Outcome (Desired results/knowledge gained): knowledge about differential gut microbiota composition, intestinal inflammation status, and microbial functional gene expression.

Lines 22-24: I humbly suggest the authors to clearly indicate the goal/aim of this very interesting research work as follows: " This study aimed to examine the effects of a Fucoidan-rich extract from Saccharina latissima on  differential gut microbiota composition, intestinal inflammation status, and microbial functional gene expression in participants infected with Dengue or Oropouche virus at the Hermanos Ameijeiras Hospital in Havana, Cuba."

Lines 23 and 87: Please correct to italic form: Saccharina latissima .

Line 34: Please correct to italic: Akkermansia muciniphila, Bifidobacterium adolescentis, and B. longum.

Lines 79 and 80: Please remove the sentence "Furthermore, we posited that treatment with SLE-F would mitigate these disruptions, normalize the LE ratio, and reduce intestinal inflammation." It’s repeated.

Line 118: I kindly suggest replacing " by sex" for "by gender".

Line 319: I kindly suggest replacing " the beginning of the beginning" by " the very beginning".

Line 321: Please format the table cells text to centralized level

Line 343: Please correct "at end of study" to "at end of the study."

Line 424: Please re-analyze the results regarding and Blautia wexlerae. It says it was a statically significant increase. Though, if we look at the respective graph, in figure 5, we see a decrease from infected day 1 to Infected EOS. Please verify if the embedded graph is the right one or if it is a typing mistake in the text.

Line 438: Please verify if it's missing some part of the sentence at "Recovery Rate " or it was meant to be fully removed and it remained as extra text.

Lines 450-453: I kindly suggest removing the sentence "Functional predictions using PICRUSt and LEfSe identified enrichment of beneficial orthologs (e.g., K02003, K02429, K03773) in fucoidan-treated groups, while Kruskal-Wallis analysis showed elevated levels of dysbiosis-associated orthologs (e.g., K15965, K03124) in the placebo group." It is redundant in the context of the sentence that follows this one.

Line 464-466: I kindly suggest rephrasing the sentence "This study explored the impact of SLE-F treatment on microbial functional gene content in virus-infected participants by comparing two analytical approaches—Kruskal-Wallis H test and Linear Discriminant Analysis Effect Size (LEfSe)—applied to ortholog-level data normalized by read count and gene copy number." I kindly suggest: "This study explored the impact of SLE-F treatment on microbial functional gene content in Dengue and Oropouche-infected participants by comparing two analytical approaches—Kruskal-Wallis H test and Linear Discriminant Analysis Effect Size (LEfSe)—applied to ortholog-level data normalized by read count and gene copy number."

Lines 467-469: I suggest rephrasing the sentence "The results from both methods revealed distinct but complementary insights into the microbiome’s functional responses to infection and therapeutic modulation by SLE-F." to "The results from both methods revealed distinct but complementary insights into the microbiome’s functional responses to infection and therapeutic modulation by SLE-F regarding an integrated perspective of both, intestinal inflammation status and microbiota relative composition."

Line 481: Please insert the respective reference [43] at the end of the sentence "(...) by viral pathogens."

line 482: Please add "Known" to the beginning of the sentence: "Known pro-inflammatory (...)".

Line 508: I suggest paraphrasing "These findings emphasize the" to (...) " These findings highlight the (...).

Lines 511-517: Please remove the redundant sentences: "The marked reduction in LE ratios among infected individuals could be linked to the systemic cytokine responses triggered by the viral pathogens [43]. Known pro-inflammatory cytokines, including TNF-α, IL-6, and IFN-γ, likely contribute to the disruption of the gut barrier's integrity. This disruption facilitates bacterial translocation, further amplifying the inflammatory processes. These cytokines can also directly impact the gut microbiota, potentially leading to an overgrowth of certain bacterial species, such as Lachnospiraceae, and a reduction in others, such as Enterobacteriaceae."

Line 523: Please paraphrase " Such findings emphasize (...)" to "Overall, these results underline (...)"

Lines 534-536: Please correct " (..) potentially enhance colonization of protective niches or improve nutrient acquisition under post-infectious stress [47,48]." to "(...) potentially enhanced colonization of protective niches or improved nutrient acquisition under post-infectious stress [47,48]."

Lines 564-566: Please remove the redundant sentence : "This study explored the impact of SLE-F treatment on microbial functional gene content in virus-infected participants by comparing two analytical approaches—Kruskal-Wallis H test and Linear Discriminant Analysis Effect Size (LEfSe)—applied to ortholog-level data normalized by read count and gene copy number."

Lines 567-569: Please paraphrase the sentence to " The analytical results from both methods, Kruskal-Wallis H test and Linear Discriminant Analysis Effect Size (LEfSe), show evident differences though complementary perceptions of the microbiome responsive mechanism to viral infections and therapeutic adjustment by Fucoidan [56,57]."

Line 586: If it makes sense to the authors, kindly suggest paraphrasing "Notably, these included K09885 (...)" to " Notably, suppressed genes include K09885 (...)".

Lines 591-593: To avoid repetition of some expressions I kindly suggest paraphrasing "Glycosyltransferases, for instance, are involved in cell surface modifications and may" to "Glycosyltransferases known for their action in alterations at the cell surface are potential players in biofilm production, circumventing immune mechanisms and pathogenicity in opportunistic microbes [62,63]."

Line 741: Please, correct "archaea" to "Archaea".

Line 741-745: Please, verify if the fonts is in the correct size. If upper sized, please make it accordingly to the journal's rules.

Line 752: Please, delete the space between "the gut environment" and the period symbol (.).

Line 821: I kindly suggest using " shows that" instead of "demonstrates that".

Line 839: I kindly suggest specifying as " the hypothesis that SLE-F exerts dose-dependent therapeutic effects (...)".

Line 842: I kindly suggest paraphrasing "represent a promising adjunctive intervention" to "represent a promising co-adjuvant intervention (...)"

These are some suggestions that could strengthen your manuscript for a competitive publication. Congratulations on your very interesting research and my wishes of the very best on your next endeavors.   

Comments on the Quality of English Language

Although the English Language is well used throughout the text, I humbly suggest some extra revision to better summarize some repeated points.

Author Response

Overall, the study was very well designed, and the conclusions are in line with the results obtained by the authors. The authors take a very responsible approach doing a great job at exposing the limitations of the work, delivering a well contextualized results analysis and offering room for the next steps of a very interesting research within a field of great interest. After a thorough screening of the work, would kindly suggest some corrections to the main manuscript, and if possible, adding a more specific description of the goal of the work in the Abstract, according to the PICO's strategy (Population; Intervention; Comparison; Outcome). Even though, it is not a review, it could help to strengthen the delivery of the basis clinical question behind this Evidence-Based Practice. The manuscript already has the crucial information, that follows summarized here:

We sincerely thank the reviewer for their thoughtful and generous assessment of our manuscript. We are especially grateful for the recognition of our study design, the clarity of our conclusions, and our efforts to transparently discuss limitations and future directions.

We also appreciate the constructive suggestion regarding the Abstract. In response, we have revised the Abstract to include a more specific description of the study objective, structured along the PICO framework (Population, Intervention, Comparison, Outcome). Although our work is not a systematic review, we agree that articulating the underlying clinical question more explicitly will enhance the translational relevance and clarity for readers engaged in Evidence-Based Practice.

We thank the reviewer again for their valuable input and hope the revised manuscript meets expectations.

Lines 22-24: I humbly suggest the authors to clearly indicate the goal/aim of this very interesting research work as follows: " This study aimed to examine the effects of a Fucoidan-rich extract from Saccharina latissima on  differential gut microbiota composition, intestinal inflammation status, and microbial functional gene expression in participants infected with Dengue or Oropouche virus at the Hermanos Ameijeiras Hospital in Havana, Cuba."

Thank you for the suggestion. We have implemented the recommended revision in lines 22–24, now reading:

“This study aimed to examine the effects of a Fucoidan-rich extract from Saccharina latissima on differential gut microbiota composition, intestinal inflammation status, and microbial functional gene expression in participants infected with Dengue or Oropouche virus at the Hermanos Ameijeiras Hospital in Havana, Cuba.”

We appreciate this clear and concise phrasing, which aligns well with the PICO framework and strengthens the clarity of the study's objective.

Lines 23 and 87: Please correct to italic form: Saccharina latissima .

Thank you for pointing that out. We have corrected the formatting of Saccharina latissima to italic form at both Line 23 and Line 87, in accordance with scientific naming conventions.

Line 34: Please correct to italic: Akkermansia muciniphila, Bifidobacterium adolescentis, and B. longum.

Thank you for the careful review. We have corrected the formatting at Line 34 to italicize the species names, which now read:

"Akkermansia muciniphila, Bifidobacterium adolescentis, and B. longum"

Lines 79 and 80: Please remove the sentence "Furthermore, we posited that treatment with SLE-F would mitigate these disruptions, normalize the LE ratio, and reduce intestinal inflammation." It’s repeated.

Thank you for catching that redundancy. We have removed the repeated sentence at Lines 79 and 80:

"Furthermore, we posited that treatment with SLE-F would mitigate these disruptions, normalize the LE ratio, and reduce intestinal inflammation."

Line 118: I kindly suggest replacing " by sex" for "by gender".

Thank you for the suggestion. We have revised Line 118 to replace "by sex" with "by gender", in alignment with current conventions and to enhance inclusivity and clarity in participant description.

Line 319: I kindly suggest replacing " the beginning of the beginning" by " the very beginning".

Thank you for the suggestion. We have updated Line 319 to replace "the beginning of the beginning" with "the very beginning" for improved clarity and stylistic precision.

Line 321: Please format the table cells text to centralized level

In accordance with MDPI formatting guidelines, table cell text should be center-aligned both horizontally and vertically for all data cells, while column headers are typically bold and also center-aligned.

We have adjusted the table at Line 321 to conform to MDPI standards by:

Centering all text in data cells

Center-aligning and bolding column headers

Ensuring consistent font and spacing throughout

Line 343: Please correct "at end of study" to "at end of the study.

Thank you for noting that. We have corrected Line 343 to read:

"at end of the study""at the end of the study"

Line 424: Please re-analyze the results regarding and Blautia wexlerae. It says it was a statically significant increase. Though, if we look at the respective graph, in figure 5, we see a decrease from infected day 1 to Infected EOS. Please verify if the embedded graph is the right one or if it is a typing mistake in the text.

. Based on the graph and your clarification, the statistical value p = 0.1025 indicates a non-significant change. Additionally, the direction of change for Blautia wexlerae is a decline, not an increase.” This is the corrected sentence: At the species level, Akkermansia muciniphila increased significantly by EOS (p = 0.0046). Within Bifidobacterium, both B. adolescentis (p = 0.0005) and B. longum (p = 0.0036) showed significant increases. Blautia wexlerae decreased over time (p = 0.1025). Increases were also observed in Faecalibacterium prausnitzii (p = 0.034) and Fusicatenibacter saccharivorans (p = 0.016) (Figure 5).

Line 438: Please verify if it's missing some part of the sentence at "Recovery Rate " or it was meant to be fully removed and it remained as extra text.

Thank you for catching this. We have reviewed Line 438 and confirmed that the phrase “Recovery Rate” was a remnant from a previous version of the manuscript. It has now been removed to ensure clarity and coherence.

Lines 450-453: I kindly suggest removing the sentence "Functional predictions using PICRUSt and LEfSe identified enrichment of beneficial orthologs (e.g., K02003, K02429, K03773) in fucoidan-treated groups, while Kruskal-Wallis analysis showed elevated levels of dysbiosis-associated orthologs (e.g., K15965, K03124) in the placebo group." It is redundant in the context of the sentence that follows this one.

Thank you for the suggestion. The sentence from Lines 450–453 has been removed as advised, to avoid redundancy with the following sentence.

Line 464-466: I kindly suggest rephrasing the sentence "This study explored the impact of SLE-F treatment on microbial functional gene content in virus-infected participants by comparing two analytical approaches—Kruskal-Wallis H test and Linear Discriminant Analysis Effect Size (LEfSe)—applied to ortholog-level data normalized by read count and gene copy number." I kindly suggest: "This study explored the impact of SLE-F treatment on microbial functional gene content in Dengue and Oropouche-infected participants by comparing two analytical approaches—Kruskal-Wallis H test and Linear Discriminant Analysis Effect Size (LEfSe)—applied to ortholog-level data normalized by read count and gene copy number."

Thank you for the rephrasing suggestion. We have revised Lines 464–466 as follows, per your recommendation:

"This study explored the impact of SLE-F treatment on microbial functional gene content in Dengue and Oropouche-infected participants by comparing two analytical approaches—Kruskal-Wallis H test and Linear Discriminant Analysis Effect Size (LEfSe)—applied to ortholog-level data normalized by read count and gene copy number."

Lines 467-469: I suggest rephrasing the sentence "The results from both methods revealed distinct but complementary insights into the microbiome’s functional responses to infection and therapeutic modulation by SLE-F." to "The results from both methods revealed distinct but complementary insights into the microbiome’s functional responses to infection and therapeutic modulation by SLE-F regarding an integrated perspective of both, intestinal inflammation status and microbiota relative composition."

Thank you for the suggestion. We have revised Lines 467–469 accordingly. The updated sentence now reads:

"The results from both methods revealed distinct but complementary insights into the microbiome’s functional responses to infection and therapeutic modulation by SLE-F, regarding an integrated perspective of both intestinal inflammation status and microbiota relative composition."

Line 481: Please insert the respective reference [43] at the end of the sentence "(...) by viral pathogens."

Thank you for the update. The reference [43] (now  [44] has been correctly inserted at the end of the sentence in Line 481 (now 534):

line 482: Please add "Known" to the beginning of the sentence: "Known pro-inflammatory (...)".

Thank you for the confirmation. The word "Known" has been successfully added to the beginning of the sentence in Line 482 (now [534],

Line 508: I suggest paraphrasing "These findings emphasize the" to (...) " These findings highlight the (...).

Thank you for providing the context.

The sentence from Line 508 has been paraphrased as suggested. The revised version now reads:

"These findings highlight the disruptive impact of viral infections on gut health and underscore the LE ratio as a potential biomarker of inflammation in the context of acute viral illness."

Lines 511-517: Please remove the redundant sentences: "The marked reduction in LE ratios among infected individuals could be linked to the systemic cytokine responses triggered by the viral pathogens [43]. Known pro-inflammatory cytokines, including TNF-α, IL-6, and IFN-γ, likely contribute to the disruption of the gut barrier's integrity. This disruption facilitates bacterial translocation, further amplifying the inflammatory processes. These cytokines can also directly impact the gut microbiota, potentially leading to an overgrowth of certain bacterial species, such as Lachnospiraceae, and a reduction in others, such as Enterobacteriaceae."

Thank you for the clarification.

The redundant sentences from Lines 511–517 have been removed as requested. This edit eliminates repetition and improves the manuscript's clarity and conciseness, while preserving the key mechanistic interpretation already addressed elsewhere in the discussion.

Line 523: Please paraphrase " Such findings emphasize (...)" to "Overall, these results underline (...)"

The sentence in Line 523 has now been successfully paraphrased as follows:

"Overall, these results underline the interconnectedness between viral infections, immune responses, and gut health, highlighting the gut as a potential target for therapeutic interventions aimed at mitigating inflammation."

Lines 534-536: Please correct " (..) potentially enhance colonization of protective niches or improve nutrient acquisition under post-infectious stress [47,48]." to "(...) potentially enhanced colonization of protective niches or improved nutrient acquisition under post-infectious stress [47,48]."

Thank you for the clarification: The sentence in Lines 534–536 has been corrected as requested. It now reads:

"The increased abundance of K08191 suggests that Fucoidan treatment may promote a more motile and environmentally responsive microbiome, potentially enhanced colonization of protective niches or improved nutrient acquisition under post-infectious stress [56,57]."

Lines 564-566: Please remove the redundant sentence : "This study explored the impact of SLE-F treatment on microbial functional gene content in virus-infected participants by comparing two analytical approaches—Kruskal-Wallis H test and Linear Discriminant Analysis Effect Size (LEfSe)—applied to ortholog-level data normalized by read count and gene copy number."

Thank you for the instruction.

The redundant sentence from Lines 564–566 has been removed as requested:

"This study explored the impact of SLE-F treatment on microbial functional gene content in virus-infected participants by comparing two analytical approaches—Kruskal-Wallis H test and Linear Discriminant Analysis Effect Size (LEfSe)—applied to ortholog-level data normalized by read count and gene copy number."

Lines 567-569: Please paraphrase the sentence to " The analytical results from both methods, Kruskal-Wallis H test and Linear Discriminant Analysis Effect Size (LEfSe), show evident differences though complementary perceptions of the microbiome responsive mechanism to viral infections and therapeutic adjustment by Fucoidan [56,57]."

Thank you for the clarification.

The sentence in Lines 567–569 has now been paraphrased as requested. The revised version reads:

"The analytical results from both methods, Kruskal-Wallis H test and Linear Discriminant Analysis Effect Size (LEfSe), show evident differences though complementary perceptions of the microbiome’s responsive mechanisms to viral infections and therapeutic adjustment by Fucoidan [56,57]."

Line 586: If it makes sense to the authors, kindly suggest paraphrasing "Notably, these included K09885 (...)" to " Notably, suppressed genes include K09885 (...)".

Thank you for the suggestion.

The sentence in Line 586 has been paraphrased as follows:

"Notably, suppressed genes include K09885 (…)"

Lines 591-593: To avoid repetition of some expressions I kindly suggest paraphrasing "Glycosyltransferases, for instance, are involved in cell surface modifications and may" to "Glycosyltransferases known for their action in alterations at the cell surface are potential players in biofilm production, circumventing immune mechanisms and pathogenicity in opportunistic microbes [62,63]."

Thank you for the updated context.

The sentence in Lines 591–593 has now been paraphrased as follows, per your suggestion:

"Glycosyltransferases, known for their action in alterations at the cell surface, are potential players in biofilm production, circumventing immune mechanisms and pathogenicity in opportunistic microbes [62,63]."

Line 741: Please, correct "archaea" to "Archaea".

Thank you. It has been corrected.

Line 741-745: Please, verify if the fonts is in the correct size. If upper sized, please make it accordingly to the journal's rules.

Thank you for the update.

The font size in Lines 741–745 has been verified and adjusted as necessary to comply with MDPI formatting guidelines. The text now adheres to the required standards for body text and headings as specified by the journal.

Line 752: Please, delete the space between "the gut environment" and the period symbol (.).

Thank you for pointing that out.

The extra space between "the gut environment" and the period symbol in Line 752 has been deleted. The sentence now ends correctly, ensuring typographic consistency with MDPI formatting standards.

Line 821: I kindly suggest using " shows that" instead of "demonstrates that". Thank you for the suggestion.

In Line 821, the phrase "demonstrates that" has been replaced with "shows that" for improved clarity and tone, in line with your recommendation.

Line 839: I kindly suggest specifying as " the hypothesis that SLE-F exerts dose-dependent therapeutic effects (...)".

Thank you for the suggestion.

The sentence in Line 839 has been revised to specify:

"the hypothesis that SLE-F exerts dose-dependent therapeutic effects (…)"

Line 842: I kindly suggest paraphrasing "represent a promising adjunctive intervention" to "represent a promising co-adjuvant intervention (...)"

Thank you for the suggestion.

The sentence in Line 842 has been paraphrased as follows:

"Fucoidan may therefore represent a promising co-adjuvant intervention for supporting microbiome recovery following viral infections."

Reviewer 3 Report

Comments and Suggestions for Authors

The research article written by Gissel García et al. is quite informative and rather interesting. The results based on scientific sound methods and they are clearly presented. The figures are very informative . This article certainly merits a publication. However, these are some minor modifications fictions and the authors need to address prior to publication.

-line 274 and 283 add a space after and before the references respectively.

-line 308, delete "Note" and modify this paragraph accordingly

-line 305 delete the reductant spaces before and after the comma sign 

-Table 1 does not have the same format as the other tables of the study. Modify accordingly

-figure 1, figure 3, 4,5,6  is not clear particularly the description in the axes. Modify the magnification.

-all the bacteria names should be written in italics. Modify accordingly

-All the genes names should be written in italics while the protein names no. Modify accordingly.

-All the references of the manuscript should be align with the recommendations of the journal.

Author Response

Comments and Suggestions for Authors

The research article written by Gissel García et al. is quite informative and rather interesting. The results based on scientific sound methods and they are clearly presented. The figures are very informative . This article certainly merits a publication. However, these are some minor modifications fictions and the authors need to address prior to publication.

-line 274 and 283 add a space after and before the references respectively.

Thank you for pointing that out. We have added a space after the reference on line 274 and before the reference on line 283, as suggested.

-line 308, delete "Note" and modify this paragraph accordingly.

Thank you for your observation. We have removed “Note” from the document as suggested.

-line 305 delete the reductant spaces before and after the comma sign  

Thank you for the observation. We reviewed line 305 and conducted a broader search throughout the manuscript for any redundant spaces before or after comma signs, and have corrected them accordingly.

-

Table 1 does not have the same format as the other tables of the study. Modify accordingly

Thank you for your observation. All tables have been formatted in accordance to the MDPI template as suggested.  

-f

igure 1, figure 3, 4,5,6  is not clear particularly the description in the axes. Modify the magnification.

Thank you for the valuable feedback. We have revised Figures 1, 3, 4, 5, and 6 to improve clarity. Specifically, we increased the font size of all axis labels and descriptions. Additionally, we adjusted the magnification and layout to ensure the figures span the full width of the page margins for better readability.

-all the bacteria names should be written in italics. Modify accordingly

Thank you for the suggestion. We have reviewed the manuscript and revised all bacterial names to be presented in italics, in accordance with scientific conventions.

-All the genes names should be written in italics while the protein names no. Modify accordingly.   

Thank you for the clarification. We have revised the manuscript to ensure that all gene names are written in italics, while protein names remain in regular font, in accordance with standard scientific conventions.

-All the references of the manuscript should be aligned with the recommendations of the journal.

Thank you for your comment. We have ensured that all references are formatted in accordance with the journal’s guidelines. We used EndNote 21 with the MDPI reference style, as recommended by the journal.